# Nonlinear Machine Learning in Warfarin Dose Prediction: Insights from Contemporary Modelling Studies

**DOI:** 10.3390/jpm12050717

**Published:** 2022-04-29

**Authors:** Fengying Zhang, Yan Liu, Weijie Ma, Shengming Zhao, Jin Chen, Zhichun Gu

**Affiliations:** 1Department of Evidence-Based Medicine and Clinical Epidemiology, West China Hospital, Sichuan University, Chengdu 610041, China; xuanyoufenglaiyi@163.com (F.Z.); sxszmwj@163.com (W.M.); shengmin39@163.com (S.Z.); 2Department of Clinical Pharmacy, Xinhua Hospital, School of Medicine, Shanghai Jiao Tong University, Shanghai 200092, China; liuyan03@xinhuame.com.cn; 3Department of Pharmacy, Ren Ji Hospital, School of Medicine, Shanghai Jiao Tong University, Shanghai 200127, China; 4Shanghai Anticoagulation Pharmacist Alliance, Shanghai Pharmaceutical Association, Shanghai 200040, China

**Keywords:** warfarin, nonlinear machine learning, algorithms, model prediction, PROBAST

## Abstract

**Objective**: This study aimed to systematically assess the characteristics and risk of bias of previous studies that have investigated nonlinear machine learning algorithms for warfarin dose prediction. **Methods:** We systematically searched PubMed, Embase, Cochrane Library, Chinese National Knowledge Infrastructure (CNKI), China Biology Medicine (CBM), China Science and Technology Journal Database (VIP), and Wanfang Database up to March 2022. We assessed the general characteristics of the included studies with respect to the participants, predictors, model development, and model evaluation. The methodological quality of the studies was determined, and the risk of bias was evaluated using the Prediction model Risk of Bias Assessment Tool (PROBAST). **Results:** From a total of 8996 studies, 23 were assessed in this study, of which 23 (100%) were retrospective, and 11 studies focused on the Asian population. The most common demographic and clinical predictors were age (21/23, 91%), weight (17/23, 74%), height (12/23, 52%), and amiodarone combination (11/23, 48%), while CYP2C9 (14/23, 61%), VKORC1 (14/23, 61%), and CYP4F2 (5/23, 22%) were the most common genetic predictors. Of the included studies, the MAE ranged from 1.47 to 10.86 mg/week in model development studies, from 2.42 to 5.18 mg/week in model development with external validation (same data) studies, from 12.07 to 17.59 mg/week in model development with external validation (another data) studies, and from 4.40 to 4.84 mg/week in model external validation studies. All studies were evaluated as having a high risk of bias. Factors contributing to the risk of bias include inappropriate exclusion of participants (10/23, 43%), small sample size (15/23, 65%), poor handling of missing data (20/23, 87%), and incorrect method of selecting predictors (8/23, 35%). **Conclusions:** Most studies on nonlinear-machine-learning-based warfarin prediction models show poor methodological quality and have a high risk of bias. The analysis domain is the major contributor to the overall high risk of bias. External validity and model reproducibility are lacking in most studies. Future studies should focus on external validity, diminish risk of bias, and enhance real-world clinical relevance.

## 1. Introduction

Warfarin is an oral anticoagulant that is widely used in clinical practice for the treatment of venous thromboembolism (VTE) and thromboembolic events associated with atrial fibrillation (AF) or heart valve replacement (HVR) [1,2]. Despite its high effectiveness and low price, its optimal use is limited by a narrow therapeutic window and the highly variable clinical response between individuals. Predicting optimal warfarin dose is influenced by many factors including demographic, clinical, environmental, and genetic factors, and multiple drug interactions, such as age, weight, height, vitamin K epoxide reductase complex subunit 1 (VKORC1) genotype, cytochrome P450 family 2 subfamily C member 9 (CYP2C9) genotype, coadministration of antiplatelet drugs, antimicrobials, nonsteroidal anti-inflammatory drugs (NSAIDs), and proton pump inhibitors, etc. [3,4,5,6,7]. Therefore, a tailored warfarin dose may help clinicians to select the appropriate dose for individualized warfarin treatment.

Many pharmacogenetic algorithms integrating clinical, demographic, and genetic variables, including linear regression (LR) models, physiologically based pharmacokinetic (PBPK) models, and nonlinear machine learning (ML) models, have been developed to predict the dose requirements of individual patients. The PBPK model can help decision making in relation to dose selection and clinical study strategies by identifying pharmacokinetic (PK) liabilities, such as poor bioavailability, high clearance, potential for drug–drug interactions (DDIs), or the need for dose adjustments in special populations [8,9,10,11]. However, due to the lack of in vivo data in patient populations, the distribution and absorption parameters could not be validated, which brought uncertainty to model parameters and outputs [8]. This hindered the practical application of the PBPK model. LR, a type of ML algorithm [12], is the most commonly used method because of its high interpretability [13]. However, the complex and nonlinear relationship between the factors mentioned above, warfarin responses, and metabolism render LR possible unsuitable to appropriately predict the warfarin maintenance dose [14,15]. Therefore, ML of nonlinear relationships between variables, warfarin response, and metabolism has been recently applied to enhance the model expression of the complicated relationship between the individual factors and warfarin dose.

Numerous published algorithm studies on warfarin dose prediction provide a plethora of information, and Asiimwe et al. have examined the methodology and the risk of bias of warfarin dose prediction algorithms as of 20 May 2020 [13]. However, this article evaluated all dose prediction algorithms and did not focus on nonlinear ML algorithms, and new evidence may exist as of 2022.

Our systematic review seeks to provide a contemporary overview of the dosing algorithms for nonlinear ML research for clinical applications. We aimed to describe the study characteristics, evaluate the methods and quality of nonlinear ML studies, discuss the performance of nonlinear ML and LR algorithms, and account for the methodological considerations of nonlinear ML.

## 2. Methods

### 2.1. Search Strategy and Selection Criteria

This manuscript has been prepared according to the Preferred Reporting Items for Systematic Reviews and Meta-Analyses (PRISMA) guidelines and the corresponding checklist (Appendix A). We performed a comprehensive search using medical subject headings and text words related to “warfarin” and “algorithm” to identify eligible studies (Appendix A). Several electronic databases were searched from the inception to December 2021: PubMed, Embase, Cochrane Library, Chinese National Knowledge Infrastructure (CNKI), China Biology Medicine (CBM), China Science and Technology Journal Database (VIP), and Wanfang Database. Additional articles were retrieved by manually searching the reference lists of relevant publications. A second search was conducted in March 2022 to identify the records published after our first search.

We selected the publications for review if they fulfilled the following inclusion criteria: (I) English or Chinese language; (II) observational or interventional studies that developed, validated, or assessed the warfarin dosing algorithm modelling with at least two predictor variables in any warfarin-treated population. The exclusion criteria included (I) LR algorithm studies; (II) physiologically based pharmacokinetic (PBPK) models for predicting the absorption, distribution, metabolism, and excretion (ADME) of synthetic or natural chemical substances in humans; (III) informal publication types (such as commentaries, letters to the editor, editorials, and meeting abstracts). Warfarin dosing algorithms were defined as computational models that are composed of predictor variables to predict the weekly or daily dose of warfarin, including dose equations, nomograms, graphs, tables, and computer programs, etc.

### 2.2. Study Selection and Extraction of Data

After excluding the records that were irrelevant to our study, two reviewers (F.Z. and Y.L.) independently screened the abstracts for potentially eligible studies. The reports were then assessed for eligibility, and any disagreements were resolved by consensus. We designed a data extraction form based on four domains: participants, predictors, model development, and model evaluation. These were adapted from the Checklist for Critical Appraisal and Data Extraction for Systematic Reviews of Prediction Modelling Studies (CHARMS) [16] and the Prediction model Risk Of Bias Assessment Tool (PROBAST) [17]. Two reviewers (F.Z. and Y.L.) extracted the data from the study reports independently and in duplicate for each eligible study, and any disagreements were resolved by consensus or a third reviewer (Z.G.). When a study reported multiple nonlinear ML algorithms, each algorithm was extracted separately. When a study reported the same nonlinear machine learning algorithm based on different variables, we extracted the best model evaluation.

### 2.3. Risk of Bias

We assessed the risk of bias in each included study by applying the PROBAST. Although the above tools focus on prediction models that consider binary or time-to-event outcomes, the author encourages the use of these tools for other outcomes and other ML techniques. Additionally, many systematic reviews of ML algorithms also used PROBAST for risk of bias assessment [18,19,20,21]. PROBAST contains 20 signaling questions from four domains (participants, predictors, outcomes, and analyses), but we considered some problems to be less relevant to nonlinear ML studies (e.g., points that assigned weights in the final model). Generally, the algorithm does not receive information regarding the assigned weights. Therefore, we did not assess the questions regarding signaling (Appendix A). Two reviewers (F.Z. and Y.L.) independently assessed the signaling questions according to the degree of compliance with the PROBAST recommendations. The disagreements were discussed until a consensus was reached. The risk-of-bias judgment for each domain was based on the answers to the signaling questions.

### 2.4. Data Synthesis

We did not conduct the formal quantitative syntheses because of the probable heterogeneity of specialties and outcomes. All extracted data were summarized and presented descriptively. All analyses were performed by Excel (version 2020).

## 3. Results

### 3.1. Study Selection

We aimed to provide a contemporary overview of dosing algorithms used in nonlinear ML research for warfarin. Figure 1 illustrates the literature search and selection process. Our electronic search retrieved 8996 records. Of these, we excluded 6134 records based on titles and abstracts. Of the 298 full-text records assessed for eligibility, 275 were excluded, and 23 studies were related to nonlinear ML.

### 3.2. General Characteristics

Table 1 summarizes the general characteristics of 23 studies. Of these, 22 studies were retrospective.

#### 3.2.1. Participants

The Asian and the International Warfarin Pharmacogenetics Consortium (IWPC) sites accounted for 78% of studies, with the top two countries being China (9/23, 39%) and the USA (2/23, 9%). The median sample size for these studies was 650 (range 148–19,060). A total of 19 (83%) studies clarified the indications. Of 23 studies, 20 (87%) studies described the target international normalized ratio (INR), which is a parameter that quantifies the coagulation activity using the prothrombin time and is used for the regular monitoring of warfarin therapy.

#### 3.2.2. Predictors

All studies explored the demographic and clinical predictors, and 14 of the 23 (61%) studies explored the genetic factors. The predictors that included at least four factors are shown in Figure 2. Age (21/23, 91%), weight (17/23, 74%), height (12/23, 52%), and amiodarone combination (11/23, 48%) were the four most common demographic and clinical predictors. CYP2C9 (14/23, 61%), VKORC1(14/23, 61%), and CYP4F2 (5/23, 22%) were the three most common genetic predictors. With respect to features selection, the top three methods were as follows: univariate analysis (8/23, 35%), expert opinion and literature review (5/23, 22%), and stepwise regression (3/23, 22%). For missing data handling, a majority of the studies (17/23, 74%) did not report or deleted directly.

#### 3.2.3. Model Development

The prediction model studies included were categorized into four types: development (with internal validation), development with external validation (same data), development with external validation (another data), and external validation only. Regarding the model type, 16 of the 23 (70%) studies were development studies, 5 of the 23 (22%) studies were development with external validation (same data) studies, 1 was a development with external validation (another data) study, and 1 was an external validation study. The single neural network model (NNM) algorithm was used for model development in 6/11 (55%) studies and was used for development with external validation (same data) in 5/11 (45%) studies. Algorithms (2 or more) (e.g., DT (decision tree), SVR (support vector regression), KNN (K-nearest neighbor), ensemble learning, and other nonlinear regression model) were used for development in 10/12 (83%) studies; 1 was used for development with external validation (another data) and 1 was used for external validation.

#### 3.2.4. Model Evaluation

Of the 23 studies, 8 (35%) studies described the R^2^ (the coefficient of determination), which is a parameter fit for accuracy measurement that represents the proportion of total interpatient variability in warfarin dose requirements, accounted for by the variables included in the algorithm. Regarding the precision (predictive accuracy), the most reported measures were MAE (the mean absolute error) in 21/23 (91%) and RMSE (the root mean square error) or the mean square error (MSE) were used in 9/23 (39%). Of the 23 studies, 19 (83%) studies described the ideal dose (the absolute prediction error between predicted dose and the actual dose was within 20% of the actual dose).

Of the included studies, the MAE ranged from 1.47 to 10.86 mg/week in development studies, from 2.42 to 5.18 mg/week in development with external validation (same data), from 12.07 to 17.59 mg/week in development with external validation (another data), and from 4.40 to 4.84 mg/week in external validation (Table 2).

### 3.3. Methods and Risk of Bias

The overall and detailed risk of bias are presented in Figure 3 and Appendix A. Specifically, 8 (35%) studies were rated to have a high risk of bias in the participant domain. The major issue in this domain is the inappropriate exclusion of participants (43%); 1 (4%) study was considered as having high risk of bias in the predictor domain; 8 (35%) studies had high risk of bias in the outcome domain; and 22 (96%) studies were judged as high risk of bias in the analysis domain (Table 3).

Thus, the analysis domain was major contributor to the overall high risk of bias. In detail, the number of participants with the outcome was unreasonable or unclear in 15 (65%) studies; 20 (87%) studies were inappropriate regarding the handling of missing data (missing data were omitted or excluded); 8 (35%) studies used univariable analyses to select the predictors. Given above, the factors contributing to the risk of bias include the inappropriate exclusion of participants (10/23, 43%), small sample size (15/23, 65%), poor handling of missing data (20/23, 87%), and incorrect method of selecting predictors (8/23, 35%) (Table 3).

## 4. Discussion

### 4.1. Major Finding

In this study, we summarized the warfarin dosing algorithms for nonlinear ML based on their general characteristics and risk of bias. The major findings were as follows: (I) Most studies on nonlinear-machine-learning-based warfarin prediction models show poor methodological quality and are at high risk of bias, mainly as a result of inappropriate exclusions, small sample size, inappropriate handling of participants with missing data, and incorrect predictor selection; (II) About 50% of the involved studies focused on the Asian population, and the included clinical variables (age, weight, height, amiodarone combination) and genetic variables (CYP2C9, VKORC1) were recognized by most studies; (III) Most studies were limited to model internal validation, and external validation was rarely used to further illustrate the generalization ability of the predictive model.

### 4.2. Risk of Bias

Overall, most studies show poor methodological quality and are at a high risk of bias. Without a rigorous evidence base, these research results may not be applicable to the clinical use and should rather guide their efforts towards improving the design and quality [45]. The following key improvements were established from our methodological review.

First, inappropriately excluding participants may result in the target population not being represented. The exclusion criteria of 7/23 (30%) studies included complications that occurred during anticoagulant therapy (thrombosis, embolism, bleeding, death result) or severe liver and kidney dysfunction after the operation, which may introduce a bias because the final study population represented a selected, lower-risk sample of the original population. This may produce biased estimates of the predictive performance of the model. Therefore, reasonable inclusion and exclusion criteria should be adopted.

Second, since EPV (the number of events per candidate predictor variable) may not be the best method for nonlinear ML in warfarin dose prediction, we applied more stringent criteria for signal identification (EPV > 200 [46]). We also found that the training set size of studies with EPV rated “N” was small, ranging from 108 to 587. This size is small for ML, making it impossible for us to judge the potential biases. Although no studies have been conducted regarding the sample size calculations for developing prediction models using nonlinear ML techniques, these studies usually require more participants than conventional statistical approaches do [47].

Third, with respect to missing data, most studies deleted or failed to provide the methods. Multiple imputation is generally preferred because it prevents biased model performance as a result of deletion or single imputation of participants’ missing data. However, in warfarin prediction models with nonlinear ML techniques, multiple imputations remain a minority. Therefore, it would be useful if algorithm developers could improve the imputation methods in their models when possible.

Forth, univariate analysis to select the predictors may be an incorrect method of predictor selection because predictors are chosen on the basis of their statistical significance as a single predictor rather than in context with other predictors [48]. Well-established predictors should be included and retained in the model, regardless of any statistical significance. Therefore, we recommend that researchers avoid the use of univariable analysis to select the candidate predictors and choose nonstatistical methods (without any statistical univariable pretesting of the associations between candidate predictors and outcome) or other methods (for example, principal component analysis and lasso regression).

### 4.3. Performance Measures

We originally aimed to evaluate all the measurement parameters of the algorithms. As reported previously, the coefficient of determination (R^2^) was the most common parameter fit for accuracy measurement, and MAE, MSE, and RMSE were the most common parameters for predictive accuracy measurement. However, as a measure of the degree of association between actual and predicted doses, R^2^ may be misleading in terms of the actual closeness of predictions to the true values. Hence, it is not a good measure of the predictive accuracy [49,50]. As the proximity distance measures between the actual and predicted doses, accuracy measure parameters (MAE, MSE, RMSE) have the following issue: 1 mg/d may be clinically more important in a small value than a similar error in a large value. For example, numerous studies have shown that the Asian populations display higher sensitivity to the anticoagulant effects of warfarin, and the overall stable dose is relatively small compared with Caucasian populations. The MAE made by the model is correspondingly smaller than that of the other races. However, this does not mean that the model is better. These values should be interpreted cautiously.

### 4.4. Linear Regression vs. Nonlinear ML in Warfarin

As a traditional and widely used oral anticoagulant, warfarin dose prediction models have been studies for a long time [13]. LR was originally used to develop a warfarin predictive model because of its simple development process and high interpretability. Gage et al. and the International Warfarin Pharmacogenetics Consortium (IWPC) have developed two representative linear warfarin prediction models based on pharmacogenomic information and clinical factors [4,51]. However, the LR method presents certain irreconcilable issues, such as poor behaviour of the nonlinear relationship between variables, which makes the LR model an inappropriate method [14,15]. With the development of science and technology, nonlinear ML has gradually emerged as a powerful technique for analysing complex analytic problems and could use nonlinear, highly interactive combinations of predictors to uncover novel patterns that may improve the predictive performance [52,53,54]. Therefore, an increasing number of researchers are beginning to develop warfarin predictive models using nonlinear ML methods to explore new possibilities.

However, identifying which algorithm performs better (either LR or nonlinear ML) is difficult, because these algorithms are derived from different racial backgrounds. Therefore, we compared nonlinear ML with LR in the same population and variables (Appendix A). Overall, the results of these studies still cannot determine which algorithm performs better. In some studies, the ML had a lower MAE than the LR, and vice versa. With these points in mind, further development and validation of warfarin dose prediction models based on nonlinear ML, and a more thorough comparison with LR, are recommended.

### 4.5. Clinical Relevance

In the included literature, we also extracted the suggestion in clinical practice (suggestion in discussion that algorithm can now be used clinically). Unfortunately, few studies underwent the description of the practical application using the developed model. Therefore, a simple guide process aiming at the population of anticoagulant therapy using warfarin might be necessary. First, surgeons should conduct a comprehensive assessment of patient conditions for the risk factors before and after procedure. Patients who are older, more fragile, and have the VKORC1 or CYP2C9 gene carrier might be at high risk of bleeding [55]. In addition, concomitant drugs to warfarin therapy may also increase the risk of bleeding, such as amiodarone, coadministration of antiplatelet drugs, antimicrobials, etc. [56]. Second, when determining warfarin doses during treatment, we suggest using externally validated dosing algorithms over an internal validated dosing algorithms or an ad hoc approach. Certainly, the dosing algorithms that have been externally validated across races, which focuses on a certain race and extends to different races via appropriate weighting or modifications, are more suitable for clinical application. Third, patients with extremely high or low doses determined by the algorithm should receive more frequent INR testing and bleeding risk assessment after surgery. Forth, we suggest enrolling patients with warfarin in a structured care process, which may include sophisticated patient tracking systems, comprehensive patient education, outcome evaluation, and quality improvement activities.

### 4.6. Study Limitations

The main limitations of this study are as follows: first, the nomenclature in the field is sometimes used in a non-standardized manner, and thus some potentially eligible studies might have included terminologies that were not captured by our search strategy. Although comprehensive, our search might have missed some studies that could have been included. Second, the benchmark that we adopted to evaluate the risk of bias (PROBAST) was designed for conventional prediction modelling studies, this has certain limitations. Third, the risk of bias entails some subjective judgment, and people with different experiences may have varying perceptions.

### 4.7. Future Research

The number of warfarin dose prediction models based on nonlinear ML is increasing every year; thus, their identification, reporting, and assessment have become even more relevant. We believe that researchers should consider the following points: first, researchers should consider the impact of different populations, indications, and valve types on the range of INR treatment range, whether the inclusion and exclusion criteria of the included participants can accurately represent the target population or not, appropriate handling of participants with missing data, and correct predictor selection methods. Second, with respect to model development, we recommend that the researchers report detailed information about modelling methods and encourage code sharing to enhance the possibility of reproducibility. Third, for the better judgement of studies, we recommend that researchers should adhere to the transparent reporting of a multivariable prediction model for individual prognosis or diagnosis (TRIPOD) statement [57]. Although TRIPOD was not explicitly developed for machine learning prediction models, all items were applicable.

## 5. Conclusions

Most warfarin dosing algorithms for nonlinear ML have been developed in Asian populations, and few algorithms have been externally validated. Most studies show poor methodological quality and are at a high risk of bias, which makes them unreliable for clinical use. Factors contributing to the risk of bias include the inappropriate exclusion of participants, small sample size, poor handling of missing data, and incorrect method of selecting predictors. Efforts to improve the design, conduct, reporting, and validation studies of warfarin dosing algorithms of nonlinear ML are necessary to boost its application in clinical practice.

## Figures and Tables

**Figure 1 jpm-12-00717-f001:**
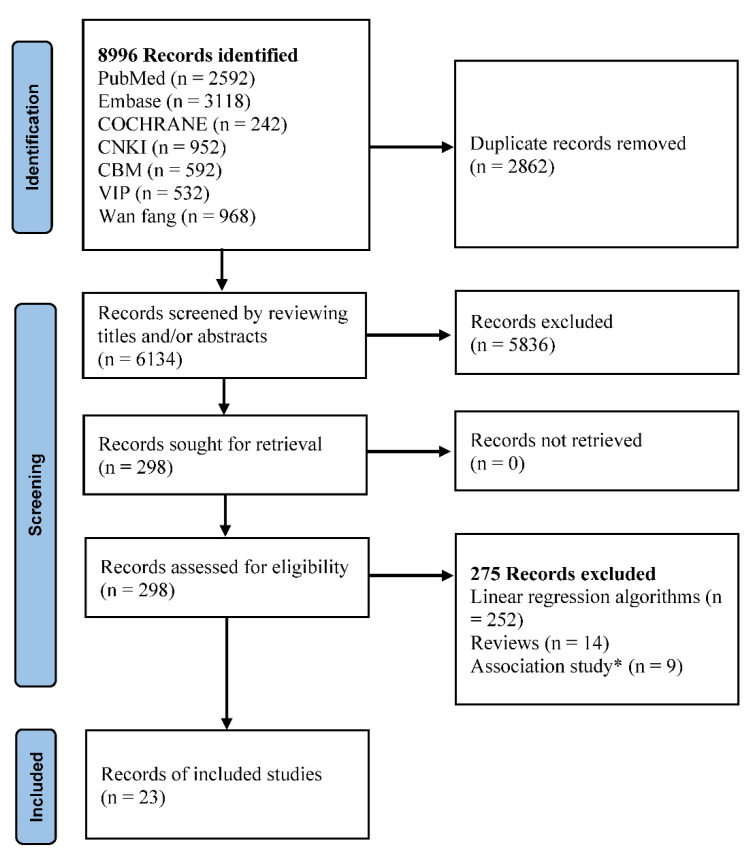
PRISMA flow chart of included studies. * The same research was translated into Chinese and English.

**Figure 2 jpm-12-00717-f002:**
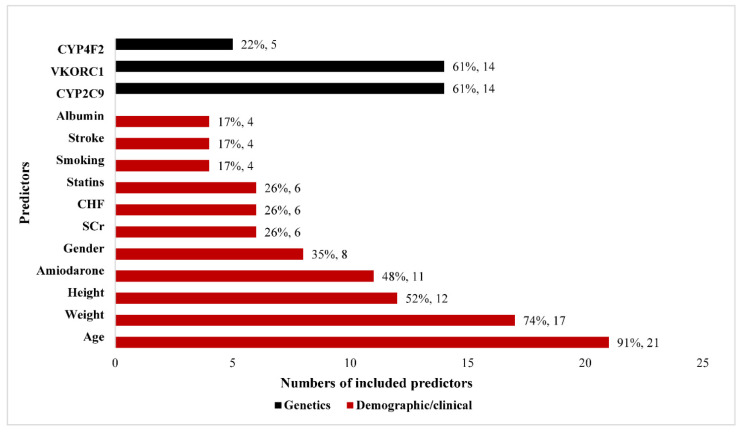
Predictors included at least 4 factors in all studies. SCr—serum creatinine; CHF—congestive heart failure.

**Figure 3 jpm-12-00717-f003:**
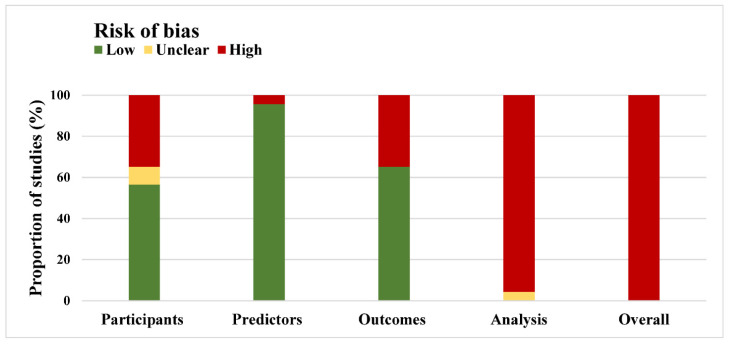
PROBAST (Prediction Model Risk of Bias Assessment Tool) risk of bias assessment for 23 studies.

**Table 1 jpm-12-00717-t001:** Summary characteristics of studies.

Study	Year	Study Type	Participants				Predictors			Model Development		Model Evaluation
			Source	Patients	Indication	Target INR *	Features	Features Selection	Missing Data Handling	Model Type	Machine Learning Algorithms ^#^	Performance Measures
Solomon [22]	2004	Retrospective	Israel	148	NA	NA	clinical	Univariate analysis	NA	Development	NNM	r
Cosgun [23]	2011	Retrospective	USA	290	NA	2.0–3.0	clinical + genetic	Univariate analysis	single imputation	Development	DT, SV, Ensemble learning	R^2^
Hu [24]	2012	Retrospective	China	587	NA	1.0–3.0	clinical	Expert opinion and literature review	NA	Development	DT, SV, KNN, Ensemble learning	MAE
Grossi [25]	2014	Retrospective	Italy	377	PE, DVT, AF, AHV, CM, Stroke, Others	2.0–4.0	clinical + genetic	Machine learning algorithm (TWIST system)	NA	Development	NNM	R^2^, MAE, ideal dose
Saleh [26]	2014	Retrospective	IWPC sites	4271	PE, DVT, AF, AHV, CM, Stroke, Others	2.0–3.0	clinical + genetic	Backward Variable Selection	Excluded	Development	NNM	R^2^, MAE, ideal dose
Zhou [27]	2014	Retrospective	China	1093	HVR	1.5–2.5	clinical	Univariate analysis, Stepwise regression	NA	Development	NNM	MAE, ideal dose
Li [28]	2015	Retrospective	China; IWPC sites	1511	HVR	1.7–3.0; 2.0–3.0	clinical + genetic	Stepwise regression	Excluded	External validation	DT, SV, NNM, Ensemble learning, Other	MAE, ideal dose
Liu [29]	2015	Retrospective	IWPC sites	4797	PE, DVT, AF, AHV, CM, Stroke, Others	2.0–3.0	clinical + genetic	Stepwise regression	Excluded	Development	DT, SV, NNM, Ensemble learning, Other	MAE, ideal dose
Alzubiedi [30]	2016	Retrospective	IWPC sites	163	PE, DVT, AF, Stroke, Others	2.0–3.0	clinical + genetic	Backward Variable Selection	NA	Development	NNM	R^2^, MAE, ideal dose
Pavani [31]	2016	NR	India	240	PE, AF, HVR	No limitation	clinical + genetic	NA	NA	Development	NNM	R^2^, MAE
Li [32]	2018	Retrospective	China	15,694	HVR	1.5–2.5	clinical	Covariance analysis, expert opinion, and literature review	NA	Development with external validation (same data)	NNM	MAE, RMSE, ideal dose
Ma [33]	2018	Retrospective	IWPC sites	5743	PE, DVT, AF, AHV, CM, Stroke, Others	1.7–3.3	clinical + genetic	Expert opinion and literature review	single imputation	Development	SV, NNM, Ensemble learning, KNN	MAE, ideal dose
Tao [34]	2018	Retrospective	China	13,639	HVR	1.5–2.5	clinical	Univariate analysis	NA	Development with external validation (same data)	NNM	MAE, MSE, ideal dose
Li [35]	2019	Retrospective	China	13,639	HVR	1.5–2.5	clinical	Univariate analysis	NA	Development with external validation (same data)	NNM	MAE, MSE, RMSE, ideal dose
Tao [36]	2019	Retrospective	China	289	NR	2.0–3.0	clinical + genetic	NA	NA	Development	NNM, SV, GP, Ensemble learning	R^2^, MAE, MSE, ideal dose
Tao [37]	2019	Retrospective	China; IWPC sites	617	PE, DVT, AF, VR, ICT, EVE, stroke	2.0–3.0; 2.0–2.5	clinical + genetic	Expert opinion and literature review	NA	Development	DT, SV, Ensemble learning	R^2^, MAE, MSE, ideal dose
Roche-Lima [38]	2020	Retrospective	USA	190	PE, DVT, AF, VR, DM2, CHF, Stroke, Others	2.0–3.0	clinical + genetic	NA	Excluded	Development	DT, SV, NNM, KNN, Ensemble learning, Other	MAE, ideal dose
Asiimwe [39]	2021	Retrospective	Uganda, South Africa	634	AF, VT, VHT	2.5–3.5; 2.0–3.0	clinical	Expert opinion and literature review	multivariate imputation	Development with external validation (another data)	DT, SV, KNN, NNM, Ensemble learning, Other	MAE, MAPE, ideal dose
Gu [40]	2021	Retrospective	China	15,108	HVR	1.5–2.5	clinical	Univariate analysis	Excluded	Development with external validation (same data)	NNM	MAE, MSE, ideal dose
Liu [41]	2021	Retrospective	China	377	PE, DVT, AF, HF, PAH, Stroke	1.5–3.0	clinical + genetic	Univariate analysis	Not imputed	Development	Ensemble learning	R^2^, MAE, MSE, RMSE, ideal dose
Ma [42]	2021	Retrospective	China	19,060	HVR	1.5–2.5	clinical	Univariate analysis	NA	Development with external validation (same data)	NNM	MAE, MSE, ideal dose
Nguyen [43]	2021	Retrospective	Korean	650	PE, DVT, HVR, VHD, Stroke, Arrhythmia, others	1.5–3.0	clinical + genetic	Recursive feature elimination	Single imputation	Development	Ensemble learning	r, MAE, RMSE, ideal dose
Steiner [44]	2021	Retrospective	IWPC sites, North and South America	7030	PE, DVT, TIA, Others	2.0–3.0; No limitation	clinical + genetic	NA	multivariate imputation	Development	DT, SV, Other	MAE, ideal dose

AF—atrial fibrillation; AHV—artificial heart valves; PE—pulmonary embolism; DVT—deep vein thrombosis; CM—cardiomyopathy; ICT—intracardiac thrombus; EVE—endovascular exclusion of aortic dissection; DM2—type 2 diabetes mellitus; CHF—congestive heart failure; HF—heart failure; VT—venous thromboembolism; VHT—valvular heart disease; PAH—pulmonary arterial hypertension; TIA—transient attack; HVR—heart valve replacement; INR—the international normalized ratio; NNM—neural network model; DT—decision tree; SV—support vector; KNN—K-nearest neighbor; GP—genetic programming; Other—other nonlinear regression model; r—coefficient of correlation; R^2^—coefficient of determination; MAE—mean absolute error; MSE—mean square error; RMSE—root mean square error. * In the article, there were different target INRs based on different indications in the same dataset, we took the minimum and maximum value of the target INRs. There were different target INRs in different datasets, we took the target INRs separately. # The nonlinear machine learning algorithms were divided into 6 categories (DT, SV, NNM, KNN, GP, ensemble learning, other nonlinear regression) based on the algorithms involved in the studies. When a study reported many subcategories in a large category, the large category was reported in the table.

**Table 2 jpm-12-00717-t002:** Performance evaluation of the included nonlinear machine learning algorithms studies.

Studies	NO. Models	Models	NO. Patients	NO. Features	MAE (mg/Week)
**Development**					
Solomon 2004	1	NNM	148	3	NR
Cosgun 2011	3	DT, SV, Ensemble learning	290	11	NR
Hu 2012	9	DT, SV, KNN, Ensemble learning	587	7	(1.47, 1.55)
Grossi 2014	1	NNM	377	14	5.72
Saleh 2014	1	NNM	4271	9	9
Zhou 2014	1	NNM	1093	11	0.08 *
Liu 2015	7	DT, SV, NNM, Ensemble learning, Other	4797	9	(8.84, 9.82)
Alzubiedi 2016	1	NNM	163	7	11.2
Pavani 2016	1	NNM	240	9	−1.97 *
Ma 2018	8	SV, NNM, KNN, Ensemble learning	5743	13	(8.31, 10.86)
Tao 2019	6	NNM, SV, GP, Ensemble learning	289	7	NR
Tao 2019	4	DT, SV, Ensemble learning	617	11	(4.73,5.36)
Roche-Lima 2020	9	DT, SV, NNM, KNN, Ensemble learning, Other	190	24	(4.73, 9.87)
Liu 2021	3	Ensemble learning	377	11	(2.98, 4.54)
Nguyen 2021	1	Ensemble learning	650	17	4.48
Steiner 2021	3	DT, SV, Other	7030	13	(8.11, 8.18)
**Development with external validation (same data)**					
Li 2018 IV	1	NNM	15,694	12	2.59
Li 2018 EV	1	NNM	15,694	12	2.68
Tao 2018 IV	1	NNM	13,639	9	4.07
Tao 2018 EV	1	NNM	13,639	9	4.22
Li 2019 IV	1	NNM	13,639	10	4.82
Li 2019 EV	1	NNM	13,639	10	5.18
Gu 2021 IV	1	NNM	15,108	8	2.58
Gu 2021 EV	1	NNM	15,108	8	2.59
Ma 2021 IV	2	NNM	19,060	8	(2.28, 3.04)
Ma 2021 EV	2	NNM	19,060	8	(2.42, 2.88)
**Development with external validation (another data)**					
Asiimwe 2021	13	DT, SV, KNN, NNM, Ensemble learning, Other	270	7	(12.07, 17.59)
**External validation**					
Li 2015	6	DT, SV, NNM, Ensemble learning, Other	1295	10	(4.41, 4.76)
Li 2015	6	DT, SV, NNM, Ensemble learning, Other	216	10	(4.40, 4.84)

NNM—neural network model; DT—decision tree; SV—support vector; KNN—K-nearest neighbor; GP—genetic programming. * The value provided in literature was not clear, and it was impossible to distinguish whether it was derived from the training set or the test set.

**Table 3 jpm-12-00717-t003:** PROBAST signaling questions in 23 included studies.

Signaling Question No.	Signaling Question	Included Studies (*n* = 23)		
		Yes or Probably Yes	No or Probably No	No Information
Participant domain		number (percentage, 95% confidence interval)		
1.1	Were appropriate data sources used, e.g., cohort, RCT, or nested case–control study data?	23 (100, 100 to 100)	0	0
1.2	Were all inclusions and exclusions of participants appropriate?	13 (57, 36 to 77)	8 (35, 15 to 54)	2 (8, 3 to 20)
**Predictor domain**				
2.1	Were predictors defined and assessed in a similar way for all participants?	23 (100, 100 to 100)	0	0
2.2	Were predictor assessments made without knowledge of outcome data?	22 (96, 87 to 100)	1 (4, 4 to 13)	0
2.3	Are all predictors available at the time the model is intended to be used?	22 (96, 87 to 100)	1 (4, 4 to 13)	0
**Outcome domain**				
3.1	Was the outcome determined appropriately?	16 (70, 51 to 89)	7 (30, 12 to 49)	0
3.3	Were predictors excluded from the outcome definition?	22 (96, 87 to 100)	1 (4, 4 to 13)	0
3.4	Was the outcome defined and determined in a similar way for all participants?	21 (91, 80 to 100)	0	2 (9, 3 to 20)
3.5	Was the outcome determined without knowledge of predictor information?	22 (96, 87 to 100)	1 (4, 4 to 13)	0
3.6	Was the time interval between predictor assessment and outcome determination?	23 (100, 100 to 100)	0	0
**Analysis domain**				
4.1	Were there a reasonable number of participants with the outcome?	8 (35, 15 to 54)	12 (52, 32 to 73)	3 (13, 1 to 27)
4.3	Were all enrolled participants included in the analysis?	23 (100, 100 to 100)	0	0
4.4	Were participants with missing data handled appropriately?	3 (13, 1 to 27)	20 (87, 73 to 100)	0
4.5	Was selection of predictors based on univariable analysis avoided?	11 (48, 27 to 68)	8 (35, 15 to 54)	4 (17, 2 to 33)
4.7	Were relevant model performance measures evaluated appropriately?	21 (91, 80 to 100)	2 (9, 3 to 20)	0
4.8	Were model overfitting and optimism in model performance accounted for?	19 (90, 78 to 100)	2 (10, 3 to 22)	0

Signaling questions 3.2, 4.2, 4.6, and 4.9 were not included (Appendix A). The risk of bias judgment for each domain was based on the answers to the signaling questions. If the answer to all signaling questions was yes or probably yes, then the domain was judged as low risk of bias. If reported information was insufficient to answer the signaling questions, these were judged as no information. If more than half of the answer to all signaling questions were judged as no information, then the domain was judged as high risk of bias, otherwise the domain was judged as unclear risk of bias. If one answer to all signaling question was answered as no or probably no, then the domain was judged as high risk of bias. After judging all the domains, we performed an overall assessment for each application of PROBAST. This tool recommends rating the study as low risk of bias if all domains had low risk of bias. If at least one domain had a high risk of bias, overall judgment was rated as high risk of bias. If the risk of bias was unclear in at least one domain and all other domains had a low risk of bias, then an unclear risk of bias was assigned.

## Data Availability

The data that support the findings of this study are available on reasonable request from the corresponding authors.

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
