# Peer review of "Nonlinear Machine Learning in Warfarin Dose Prediction: Insights from Contemporary Modelling Studies"

_jpm, 2022, doi:10.3390/jpm12050717_

Round 1

Reviewer 1 Report

Zhang et al. nicely presented an assessment of the characteristics and biases of the published machine learning algorithms for warfarin dose prediction. This is very important, especially for the pharmacogenomics community, as there are so many algorithms models published and guidance to appraise and potentially use them in developing clinical protocols is crucial. Authors nicely present their conclusions and provide rough advice for assessment and future studies. I would like to ask the authors for comment in the discussion section about the utility and if machine learning could provide predictions for other populations based on results on specific people (e.g., Asians) using the appropriate modifications, weighting, etc.

Author Response

Major comments:

Zhang et al. nicely presented an assessment of the characteristics and biases of the published machine learning algorithms for warfarin dose prediction. This is very important, especially for the pharmacogenomics community, as there are so many algorithms models published and guidance to appraise and potentially use them in developing clinical protocols is crucial. Authors nicely present their conclusions and provide rough advice for assessment and future studies. I would like to ask the authors for comment in the discussion section about the utility and if machine learning could provide predictions for other populations based on results on specific people (e.g., Asians) using the appropriate modifications, weighting, etc.

Response: We thank the Reviewer for the positive comments about our study. As reviewer stated, clinical utility of warfarin prediction models is a critical issue to external validity. We accordingly have added “clinical relevance section” in the discussion section as follows: In the included literature, we also extracted the suggestion in clinical practice (suggestion in discussion that algorithm can now be used clinically). Unfortunately, few studies underwent the description of the practical application using the developed model. Therefore, a simple guide process aiming at the population of anticoagulant therapy using warfarin might be necessary. First, surgeons should conduct a comprehensive assessment of patient conditions for the risk factors before and after procedure. Patients with older, fragile, VKORC1 or CYP2C9 gene carrier might be at high-risk of bleeding[1]. In addition, concomitant drugs to warfarin therapy may also increase the risk of bleeding, such as amiodarone, antiplatelet drugs, antimicrobials, NSAIDs, selective serotonin reuptake inhibitors (SSRIs), mirtazapine, loop diuretics, etc.[2, 3]. When determining warfarin doses during treatment, we suggest using externally validated dosing algorithms over an internal validated dosing algorithms or an ad hoc approach. Certainly, the dosing algorithms that have been externally validated across races, which focuses on a certain race and extends to different races via appropriate weighting or modifications, are more suitable for clinical application. Second, Patients with extremely high- or low-doses determined by algorithm should receive more frequent INR testing and risk assessment after surgery. Third, we suggest enrolling patients with warfarin in a structured care process, which may include sophisticated patient tracking systems, comprehensive patient education, outcome evaluation, and quality improvement activities.

References:

[1]Verhoef T I, Redekop W K, Daly A K, et al. Pharmacogenetic-guided dosing of coumarin anticoagulants: algorithms for warfarin, acenocoumarol and phenprocoumon. Br J Clin Pharmacol. 2014;77(4): 626-641.

[2]Flaker G, Lopes R D, Hylek E, et al. Amiodarone, anticoagulation, and clinical events in patients with atrial fibrillation: insights from the ARISTOTLE trial. J Am Coll Cardiol. 2014;64(15): 1541-1550.

[3]Wang M, Zeraatkar D, Obeda M, et al. Drug-drug interactions with warfarin: A systematic review and meta-analysis. Br J Clin Pharmacol. 2021;87(11): 4051-4100.

Thank you for your time and consideration.

With best regards,

Sincerely yours,

Reviewer 2 Report

Dear Editor,

The authors of the manuscript entitled "Non-linear Machine Learning in Warfarin Dose Prediction: a systematic review" aimed to systematically assess the characteristics and risk of bias of previous studies that investigated non-linear machine learning algorithms for warfarin dose prediction. The methodological quality of the studies was determined by reviewing the main scientific databases and the risk of bias was assessed using the PROBAST system. Of a total of 8996 studies, 23 were assessed in this study. The most common demographic and clinical predictors were age (21/23, 91%), weight (17/23, 74%), height (12/23, 52%), and amiodarone combination (11/23, 48%), whereas CYP2C9 (14/23.61%), VKORC1 (14/23.61%), and CYP4F2 (5/23.22%) were the most common genetic predictors. Among the included studies, the MAE ranged from 1.47 to 10.86 mg/week in the model development studies. All studies were rated at high risk for bias. The analysis range contributed most to the overall high risk of bias. They concluded, "Most studies of nonlinear ML-based warfarin prediction models have poor methodologic quality and are at high risk for bias."

The topic is interesting. The authors made extreme efforts to work up the subject.

However, there are some drawbacks in the manuscript that hinder its immediate publication. But in detail:

First of all, the lack of line and page numbering complicates the review of the manuscript. Overall, the manuscript is very long. It would be better to limit it to essential points, as there is a risk of losing the actual core message.

Language: acceptable to good, but needs revision. Some abbreviations are not explained when they first appear in the text: e.g. ML in the abstract is not explained until it appears in the introduction. Also e.g. mg/w is explained in the text but not in table 2. Furthermore, e.g. heart valve replacement is abbreviated as HVE, but should be HVR.

Abstract: The abstract is somehow difficult and needs to be better structured; e.g., "Factors contributing to the risk of bias include inappropriate exclusion of participants, small sample size, poor handling of missing data, and incorrect method of predictors selected." It sounds like a finding and therefore should come in the results section. As the author, you have of course gone very deep into the subject. However, the reader should be able to read the abstract in a relaxed manner, feel addressed and be able to understand it immediately.

Introduction: acceptable

Methods: Good structured. However, statistics are unfortunately not included. The manuscript is very similar in structure and message to the work of Asiimwe [Asiimwe et al. Warfarin dosing algorithms: A systematic review. Br J Clin Pharmacol. 2021;87(4): 1717-1729.]. He also largely lacked statistics, but at least provided medians. Overall, the statements and conclusions supported by the results may be therefore questionable.

Results: The results are only descriptive and as already mentioned without statistics. For a review with the core statement "...poor methodological quality and are at a high risk of bias..." one would expect somewhat more but a descriptive presentation, well aware that this is difficult.

Discussion: acceptable

Literature: Two different literature references are presented. This is confusing and should be changed.

Author Response

Detailed Responses to Reviewer #2

Major comments:

1. Dear Editor, the authors of the manuscript entitled "Non-linear Machine Learning in Warfarin Dose Prediction: a systematic review" aimed to systematically assess the characteristics and risk of bias of previous studies that investigated non-linear machine learning algorithms for warfarin dose prediction. The methodological quality of the studies was determined by reviewing the main scientific databases and the risk of bias was assessed using the PROBAST system. Of a total of 8996 studies, 23 were assessed in this study. The most common demographic and clinical predictors were age (21/23, 91%), weight (17/23, 74%), height (12/23, 52%), and amiodarone combination (11/23, 48%), whereas CYP2C9 (14/23.61%), VKORC1 (14/23.61%), and CYP4F2 (5/23.22%) were the most common genetic predictors. Among the included studies, the MAE ranged from 1.47 to 10.86 mg/week in the model development studies. All studies were rated at high risk for bias. The analysis range contributed most to the overall high risk of bias. They concluded, "Most studies of nonlinear ML-based warfarin prediction models have poor methodologic quality and are at high risk for bias." The topic is interesting. The authors made extreme efforts to work up the subject. However, there are some drawbacks in the manuscript that hinder its immediate publication. But in detail: First of all, the lack of line and page numbering complicates the review of the manuscript. Overall, the manuscript is very long. It would be better to limit it to essential points, as there is a risk of losing the actual core message.

Response: We thank the reviewer for the constructive suggestions on this issue. As suggested, we have added line and page numbers in the revised manuscript and invited a native speaker to further polish the language. At the same time, the revised main text was shortened from 3942 to 3347 words. The revised statements are marked with red in the submitted manuscript and mainly focus on the results and discussion section as follows:

---In the Results section (from 417 to 184 words): Methods and risk of bias: Overall and detailed risk of bias are presented in Figure 3 and Table S4. Specifically, 8 (35%) studies were rated to have a high risk of bias in the participant domain. Major issue in this domain is inappropriately exclusion of participants (43%); one (4%) study was considered as having high risk of bias in the predictor domain; 8 (35%) studies had high risk of bias in the outcome domain; while 22 (96%) studies were judged as high risk of bias in the analysis domain (Table 3). Thus, the analysis domain was major contributor to the overall high risk of bias. In detail, the number of participants with the outcome was unreasonable or unclear in 15 (65%) studies; 20 (87%) studies were inappropriate regarding the handling of missing data (missing data were omitted or excluded); 8 (35%) studies used univariable analyses to select the predictors. Given above, factors contributing to the risk of bias include the inappropriately exclusion of participants (10/23, 43%), small sample size (15/23, 65%), poor handling of missing data (20/23, 87%), and incorrect method of selected predictors (8/23, 35%) (Table 3).

---In the Discussion section (from 525 to 401 words): Risk of bias: Overall, most studies show poor methodological quality and are at a high risk of bias. Without a rigorous evidence base, these research results may not be applicable to the clinical use and should rather guide their efforts towards improve the design and quality. The following key improvements were established from our methodological review.

First, inappropriately excluding participants may result in the target population not being represented. The exclusion criteria of 7/23 (30%) studies included complications that occurred during anticoagulant therapy (thrombosis, embolism, bleeding, death result) or severe liver and kidney dysfunction after the operation, which may introduce a bias because the final study population represented a selected lower-risk sample of the original population. This may produce biased estimates of the predictive performance of the model. Therefore, reasonable inclusion and exclusion criteria should be adopted.

Second, Since EPV (the number of events per candidate predictor variable) may not be the best method for non-linear ML in warfarin dose prediction, we applied more stringent criteria for signal identification (EPV>200). We also found that the training set size of studies with EPV rated "N" was small, ranged from 108 to 587. This size is small for ML, making it impossible for us to judge the potential biases. Although no studies have been conducted regarding the sample size calculations for developing prediction models using non-linear ML techniques, these studies usually require more participants than do conventional statistical approaches.

Third, with respect to missing data, most studies deleted or failed to provide the methods. Multiple imputation is generally preferred because it prevents biased model performance as a result of deletion or single imputation of participants’ missing data. However, in warfarin prediction models with non-linear ML techniques, multiple imputations remain as a minority. Therefore, it would be useful if algorithm developers could improve the imputation methods in their models when possible.

Forth, univariate analysis to select the predictors may be an incorrect method of predictor selection because predictors are chosen on the basis of their statistical significance as a single predictor rather than in context with other predictors. Well-established predictors should be included and retained in the model, regardless of any statistical significance. Therefore, we recommend researchers should avoid the use of univariable analysis to select the candidate predictors and choose non-statistical methods (without any statistical univariable pretesting of the associations between candidate predictors and outcome) or other methods (for example, principal components’ analysis and lasso regression).

---Other revised statements are marked with red in manuscript.

2. Language: acceptable to good, but needs revision. Some abbreviations are not explained when they first appear in the text: e.g., ML in the abstract is not explained until it appears in the introduction. Also e.g., mg/w is explained in the text but not in table 2. Furthermore, e.g., heart valve replacement is abbreviated as HVE, but should be HVR.

Response: We thank the Reviewer for the positive comments about our study. As suggested, we have revised relevant content in the submitted manuscript.

3. Abstract: The abstract is somehow difficult and needs to be better structured; e.g., "Factors contributing to the risk of bias include inappropriate exclusion of participants, small sample size, poor handling of missing data, and incorrect method of predictors selected." It sounds like a finding and therefore should come in the results section. As the author, you have of course gone very deep into the subject. However, the reader should be able to read the abstract in a relaxed manner, feel addressed and be able to understand it immediately.

Response: We thank the Reviewer for the positive comments about our study. We accordingly have added “Factors contributing to the risk of bias include inappropriately exclusion of participants (10/23, 43%), small sample size (15/23, 65%), poor handling of missing data (20/23, 87%), and incorrect method of selected predictors (8/23, 35%).” in the results section of abstract and have made corresponding changes in the results and discussion section of text.

4. Methods: Good structured. However, statistics are unfortunately not included. The manuscript is very similar in structure and message to the work of Asiimwe [Asiimwe et al. Warfarin dosing algorithms: A systematic review. Br J Clin Pharmacol. 2021;87(4): 1717-1729.]. He also largely lacked statistics, but at least provided medians. Overall, the statements and conclusions supported by the results may be therefore questionable. AND 5. Results: The results are only descriptive and as already mentioned without statistics. For a review with the core statement "...poor methodological quality and are at a high risk of bias..." one would expect somewhat more but a descriptive presentation, well aware that this is difficult.

Response: We thank the Reviewer for the positive and constructive suggestion about our study. As reviewer stated, statistics are not included. Although we did not conduct the formal quantitative syntheses because of the probable heterogeneity of specialties and outcomes, certain statements, such as medians, have provided as follows: The median value for these studies was 650 (range 148–19060). In addition, the proportional description, including general characteristics and risk of bias, has been provided. Certainly, it is a pity that quantitative synthesis cannot be carried out. Some studies have tried to assess the relationship between trial characteristics and outcome measures, which also provides important clues for our future research. Thus, the main purpose of article is to provide a contemporary overview of the dosing algorithms for non-linear machine learning research for clinical applications, including describe the study characteristics, evaluate the methods and quality of non-linear machine learning studies, and the methodological considerations of non-linear machine learning. Therefore, we summarized the characteristics and evaluate risk of bias, and draw corresponding conclusions. Thanks for your kindly understanding about this issue.

6. Literature: Two different literature references are presented. This is confusing and should be changed.

Response: We thank the Reviewer for the positive and constructive suggestion about our study. we have revised relevant content in the submitted manuscript.

Thank you for your time and consideration.

With best regards,

Sincerely yours,

Jin Chen, Department of Evidence-Based Medicine and Clinical Epidemiology, West China Hospital, Sichuan University, Chengdu, China. Email: [email protected].

OR

Zhi-Chun Gu, MD, Department of Pharmacy, Ren Ji Hospital, Shanghai Jiao Tong University School of Medicine, Shanghai 200127, China; Email: [email protected].

Reviewer 3 Report

The authors treat an interesting topic about the personalization of walfarin dosage on the basis of non-linear machine learing algorithms.

It is an interesting study but in introduction section author shour cite more clinical impact of drugs and polypharmacy (10.1007/s40520-018-0893-1) to eplicitate the importance of a correct and appropriate dosage.

Author Response

Major comments:

The authors treat an interesting topic about the personalization of warfarin dosage on the basis of non-linear machine learning algorithms. It is an interesting study but in introduction section author should cite more clinical impact of drugs and polypharmacy (10.1007/s40520-018-0893-1) to elucidate the importance of a correct and appropriate dosage.

Response: We thank the Reviewer for the positive and constructive suggestion about our study. As suggested, we have revised relevant content to the introduction section as follows: Despite its high effectiveness and low price, its optimal use is limited by the narrow therapeutic window and highly variable clinical response between individuals. Predicting optimal warfarin dose is influenced by many factors including demographic, clinical, environmental and genetic factors, and multiple drug interactions, such as age, weight, height, vitamin K epoxide reductase complex subunit 1 (VKORC1) genotype, cytochrome P450 family 2 subfamily C member 9 (CYP2C9) genotype, coadministration of antiplatelet drugs, antimicrobials, non-steroidal anti-inflammatory drugs (NSAIDs), and proton pump inhibitors, etc. [3-7]. Therefore, a tailored warfarin dose may help clinicians to select the appropriate dose for individualized warfarin treatment.

References:

[3]Wang M, Zeraatkar D, Obeda M, et al. Drug-drug interactions with warfarin: A systematic review and meta-analysis. Br J Clin Pharmacol. 2021;87(11): 4051-4100.

[4]Loebstein R, Yonath H, Peleg D, et al. Interindividual variability in sensitivity to warfarin-Nature or nurture? Clinical Pharmacology & Therapeutics. 2001;70(2): 159-164.

[5]Klein T E, Altman R B, Eriksson N, et al. Estimation of the warfarin dose with clinical and pharmacogenetic data. The New England journal of medicine. 2009;360(8): 753-764.

[6]Bourgeois S, Jorgensen A, Zhang E J, et al. A multi-factorial analysis of response to warfarin in a UK prospective cohort. Genome Med. 2016;8(1): 2.

[7]Spina E, Barbieri M A, Cicala G, et al. Clinically relevant drug interactions between newer antidepressants and oral anticoagulants. Expert Opin Drug Metab Toxicol. 2020;16(1): 31-44.

Thank you for your time and consideration.

With best regards,

Sincerely yours,

Reviewer 4 Report

This is a novel article that features a systematic review and is aimed at assessment of the characteristics and risk of bias of the published studies that investigated non-linear machine learning algorithms for warfarin dose prediction.

The conclusions are vague and they basically repeat the same sentences time and time again.

Please, change the word Ancient in the subsection: Linear regression VS. non-linear ML in warfarin

Author Response

Major comments:

---This is a novel article that features a systematic review and is aimed at assessment of the characteristics and risk of bias of the published studies that investigated non-linear machine learning algorithms for warfarin dose prediction. The conclusions are vague and they basically repeat the same sentences time and time again.

Response: We thank the Reviewer for the positive and constructive suggestion about our study. As suggested, we have revised relevant content to the abstract and conclusions section as follows.

In the Abstract: Conclusion: Most studies on non-linear ML based warfarin prediction models show poor methodological quality and are at a high risk of bias. Factors contributing to the risk of bias include the inappropriately exclusion of participants, small sample size, poor handling of missing data, and incorrect method of selected predictors. External validity and model reproducibility are lacking in most studies. Future studies should focus on external validity, diminish risk of bias and enhance real world clinical relevance.

In the Conclusions: Most warfarin dosing algorithms for non-linear ML have been developed in the Asian populations, and few algorithms have been externally validated. Most studies show poor methodological quality and are at a high risk of bias, which makes them unreliable for clinical use. Factors contributing to the risk of bias include the inappropriately exclusion of participants, small sample size, poor handling of missing data, and incorrect method of selected predictors. Efforts to improve the design, conduct, reporting, and validation studies of warfarin dosing algorithms of non-linear ML are necessary to boost its application in clinical practice.

---Please, change the word Ancient in the subsection: Linear regression VS. non-linear ML in warfarin.

Response: We thank the Reviewer for the constructive suggestion about this issue. As suggested, we have revised relevant content as follows: As a traditional and widely used oral anticoagulant, warfarin dose prediction models have been studies for a long time.

Thank you for your time and consideration.

With best regards,

Sincerely yours,

Round 2

Reviewer 4 Report

Thank you for addressing the comments

Author Response

Thanks for your positive comments about our work.